

**The application of Budyko framework to irrigation districts in China**
**under various climatic conditions**
Hang Chen[1]; Zailin Huo[2*]; Lu Zhang[3]; Jing Cui[4]; Yingying Shen[4]; Zhenzhong
Han[4]
1 College of Civil Engineering and Architecture, Guangxi University, Nanning 530000, PR
China
2 Centre for Agricultural Water Research in China, China Agricultural University, Beijing
100083, PR China
3 CSIRO Land and Water, Canberra, ACT 2601, Australia
4 China Irrigation and Drainage Development Centre, Beijing 100054, PR China
**\*Corresponding author:**
Prof. Zailin Huo
Centre for Agricultural Water Research in China
China Agricultural University
No.17 Qinghua East Road, Haidian, Beijing, 100083, PR China
Phone: +86-10-62736762; Fax: +86-10-62736762; Email: huozl@cau.edu.cn





**Abstract**
Budyko's framework has been widely used to study basin-scale water balance. In this
study, we focus on the extended application of Fu's equation (one formulation of the Budyko-
type curves) to 371 large irrigation districts in China over a period of 2010-2017. Water balance
method was used to estimate actual evapotranspiration ($ET$) in the irrigated areas. Considering
the contribution of shallow groundwater to $ET$, the water availability in the Budyko framework
defined as equivalent precipitation ($P_e$) for irrigation areas is the sum of irrigation water ($I$),
precipitation ($P$) and groundwater evaporation ($ET_{gw}$). Results showed that the relationships
between evapotranspiration ($ET$), water availability ($P_e$) and energy supply ($ET_0$) can be
accurately described by the Budyko's curves. The Fu's equation performed better in humid and
semi-humid regions than arid and semi-arid regions. The comparison between $\partial ET/\partial P_e$ and
$\partial ET/\partial ET_0$ confirmed the relative effect of water availability and energy supply on $ET$
according to the variation of climatic conditions. The optimal values of Budyko parameter ω
for each irrigation district was obtained with multi-annual data using least square method.
Normalized Difference Vegetation Index (NDVI) and soil property (denoted by the proportion
of clay and sand) were selected to develop empirical equation for parameter ω using multiple
linear regression analysis method. This study showed that the Budyko framework can be
extended to irrigation areas and provide useful information on evapotranspiration to assist in
water management in irrigation areas.
**Keywords:** Budyko hypothesis; irrigation districts; NDVI and soil property; empirical
equation
**1. Introduction**
Quantifying the partitioning of precipitation ($P$) at land surface into evapotranspiration
($ET$) and runoff ($R$) is of great importance in hydrology and water resources management.



Serving as an effective tool to assess the partitioning, the Budyko framework proposed by
Budyko (1974) has been widely used in global and regional scales within the past several
decades (Caracciolo et al., 2018; Gerrits et al., 2009; Moussa and Lhomme, 2016; Roderick
and Farquhar, 2011; Troch et al., 2013; Wang and Hejazi, 2011). The original Budyko
formulation without parameters was assumed to be used in large basins at time scale
significantly longer than 1 year (Gentine et al., 2012; Roderick and Farquhar, 2011), in which
the evapotranspiration is dependent on the balance between energy supply and water
availability. With the emergent deviation of measured data from the Budyko curve, however,
more attention has been recently focused on the influence of catchment features or scales
analysis on $ET$ (Donohue et al., 2007). In this context, many studies subsequently derived
Budyko-type formulations are parametric. For example, by building on the water balance for
soil vadose zone,  Milly (1994) developed one-parameter model to evaluate the dependence of
water balance on water storage variation. Using the field observation data, Choudhury (1999)
evaluated the performance of an empirical Budyko-based equation for estimating annual $ET$
with precipitation, net radiation, and an adjustable parameter $n$, which was found to be fairly
effective in explaining the $ET$ variation. On the basis of previous works, Zhang et al. (2001)
introduced a plant-available water coefficient (w) with a range of 0.5-2.0 to assess the long-
term average effect of vegetation changes on catchment evapotranspiration. Based on a
generalization of proportionality hypothesis of the Soil Conservation Service model, Wang and
Tang (2014) derived a single-parameter Budyko-type model for mean annual water balance.
The equations mentioned above work better to control the partition of water availability and
determine the shape of Budyko curves by incorporating the influence of specific catchments
characteristics on regional hydrological cycles (Xiong and Guo, 2012; Xu et al., 2013).



Among the equations proposed for the Budyko framework, Fu's equation (Fu, 1981) with

an empirical parameter ω introduced has been used worldwide since recommended by Zhang

et al. (2004):

$$\frac{ET}{P} = 1 + \frac{ET_0}{P} - [1 + (\frac{ET_0}{P})^\omega]^{1/\omega} \tag{1}$$

where $ET$ is the actual evapotranspiration, mm; $ET_0$ is the potential evapotranspiration, mm; $P$

is the precipitation, mm; ω is a dimensionless empirical parameter that determines the shape of

the Budyko curve. Interestingly, parameter ω was found to be closely related to parameter $n$ of

Choudhury's method through $\omega = n + 0.72$ (Yang et al., 2008).

Previous studies showed that the variation of parameters in the Budyko-type equations

can be influenced by catchment characteristics (Berghuijs et al., 2014; Gentine et al., 2012;

Potter et al., 2005; Shao et al., 2012; Williams et al., 2012; Yang et al., 2009; Yokoo et al.,

2008). The consideration of vegetation can improve the performance of the Budyko framework

when extended into small regions (Donohue et al., 2007). Soil texture affects the vegetation

growth through the water-holding capacity (Porporato et al., 2004; Yang et al., 2007). The

shallow groundwater that contributes to $ET$, especially in arid and semi-arid areas, is taken as

potential water resource for water availability (Istanbulluoglu et al., 2012; Wang, 2012; Wang

and Zhou, 2016); and topography embedded its influence in hydrological cycle by regulating

the partition of precipitation into runoff (Yao et al., 2016; Zhang et al., 2004). For agricultural

areas with increasing food demand, the wide range of human activities including irrigation

events have already altered land cover and regional $ET$ (Xing et al., 2018), and half of the

irrigation water is consumed through evaporation globally (Jackson et al., 2001). In China, 40%

of total arable lands rely on irrigation events (Jin and Young, 2001). The application of water

diversion for irrigation districts has transferred the local natural hydrological processes to a

new water balance. For the areas with shallow groundwater, the groundwater evaporation also





contributes to crop growing. Thus, the hydrological impact of irrigation events and shallow
groundwater should be taken into account while the Budyko framework is used to regulate
precipitation partitioning in irrigation districts. In this study, using the data collected from 371
large-sized irrigation districts under various climatic conditions across China, we aim 1) to
assess the performance of Fu's equation in the irrigation districts in China by including external
water resource into water availability; then 2) to evaluate which factors affect the variation of
Fu's parameter ω; and 3) to develop an empirical relationship for estimating model parameter
ω using readily available data from irrigation areas in China.
**2. Materials and methods**
**2.1 Study area and data processing**

A total of 371 large-sized artesian diversion irrigation districts with designed irrigation

area covering from 200 to 10000 km$^2$ across China were selected in this study (Fig.1A). The
irrigation areas are classified as arid, semi-arid, semi-humid and humid areas according to the
values of aridity index (Tab.1). The information about each irrigation district including the
location of centre (longitude and latitude), irrigation area, groundwater depth, annual gross
irrigation water ($I$), and irrigation water use efficiency ($\eta$) over the period of 2010-2017 were
measured and provided by China Irrigation and Drainage Development Centre. The detailed
measurement processes and methods of net irrigation water and irrigation water use efficiency
are shown in Fig.B1 and Tab.C1. Daily precipitation and monthly meteorological data
including wind speed, air temperature, and relative humidity from weather stations on or
around the selected irrigation districts covering the same period were downloaded from China
meteorological data network (http://data.cma.cn/) (Fig.1A). The potential evapotranspiration
($ET_0$) is estimated as suggested by Shuttleworth (1993):

$$ET_0 = \frac{\Delta}{\Delta+\gamma}(R_n - G) + \frac{\gamma}{\Delta+\gamma}0.26(1 + 0.54u_2)(e_s - e_a) \qquad (2)$$



where $\Delta$ is the slope of the saturation vapor pressure curve, $\text{kPa}°\text{C}^{-1}$; $\gamma$ is the psychometric
constant (approximately $0.067 \text{ kPa}°\text{C}^{-1}$); $R_n$ and $G$ are the net radiation and ground heat,
$\text{MJm}^{-2}\text{day}^{-1}$; $u_2$ is the mean wind speed at 2 m height above the ground, $\text{m/s}$; $e_s$ is the
saturated water vapor pressure and $e_a$ is the actual saturated water vapor pressure, kPa. The
estimated mean monthly values of $ET_0$ were accumulated into annual values.

Tab.1 Climatic diversion according to aridity index

| Aridity condition | Aridity index | Aridity condition | Aridity index |
|---|---|---|---|
| Humid | $ET_0/P \leq 1.0$ | Semi-humid | $1.0 < ET_0/P \leq 1.5$ |
| Semi-arid | $1.5 < ET_0/P \leq 4.0$ | Arid | $ET_0/P > 4.0$ |

Due to the lack of shape maps of each irrigation district, the circles with same area as
irrigation districts were used to locate the irrigation districts on map in present study. The
values of NDVI (Normalized Difference Vegetation Index) were extracted from MOD13A1
products with spatial-temporal resolution of 500 m and 16 d (Fig.1B), which were available to
download from the NASA Data Centre at https://reverb.echo.nasa.gov. All original images
were pre–projected in a Universal Transverse Mercator (UTM) projection by Modis
Reprojection Tool (MRT). The Digital Elevation Model (DEM) data with a spatial resolution
of 1 km were downloaded at http://srtm.csi.cgiar.org/. The distribution map of soil texture
denoting the proportion of sand and clay was provided by Data Centre for Resources and
Environmental Sciences, Chinese Academy of Sciences (RESDC) (http://www.resdc.cn)
(Fig.1CD).

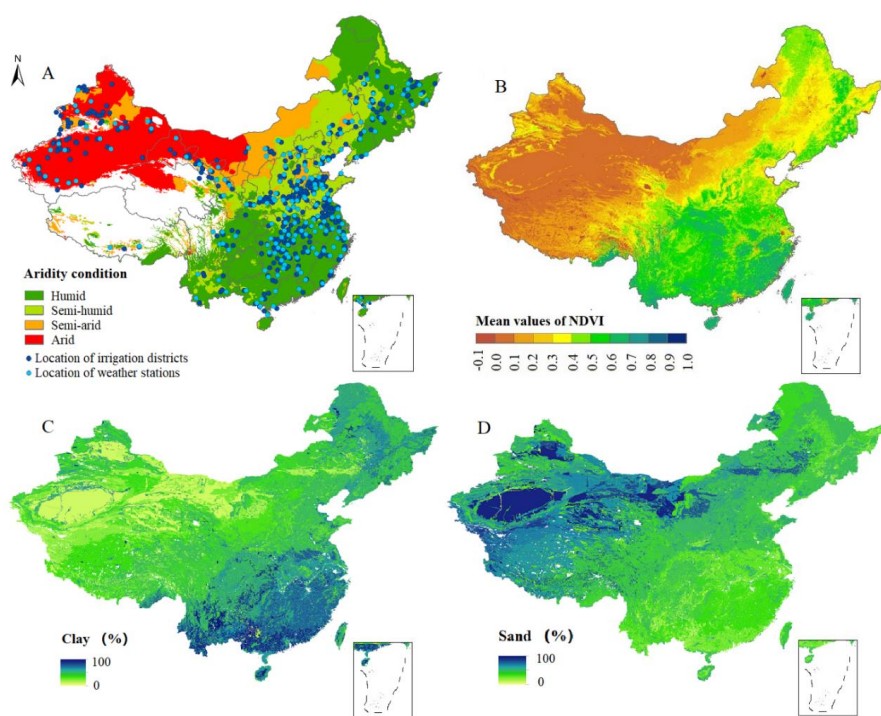


Fig.1 Location of the selected irrigation districts in this study (A) and the distribution map of
mean values of NDVI (B), proportion of clay (C) and sand (D) in China
**2.2 Theoretical framework**

Precipitation is taken as water availability in original Budyko framework when applied in

natural and closed catchments (Budyko, 1974). When extended into areas with irrigation
activities, however, precipitation is no longer the only water source for evapotranspiration. For
irrigation districts in arid and semi-arid regions, the agricultural productivity relies heavily on
the irrigation events. For humid and semi-humid regions, most of the concentrated rainfall
leaves irrigation districts by runoff and is not consumed for crop growth. A certain amount of
water is still employed to achieve the optimal agricultural productivity. In addition to providing
water for crop growth in arid and semi-arid regions, the irrigation events are also responsible
to offset water deficit caused by the unevenly distributed rainfall in humid and semi-humid





regions. Thus, the irrigation water should be included in water availability when the Budyko
framework is applied in agricultural irrigation districts (Han et al., 2011). In addition, the
groundwater evaporation consumed for crop growth, especially in arid and semi-arid areas with
shallow groundwater depth, contributes to the water availability. According to our previous
study (Chen et al., 2020), the modified Aver'yanov's phreatic equation is applicable to estimate
the groundwater consumption for the irrigation districts with shallow groundwater depths
lower than 3 m:

$$ET_{gwi} = K_C \times E_{pani} \times (1 - \frac{H_i}{H_{max}})^n \tag{3}$$

where $K_C$ is crop coefficient related to the crop growth and root length (Cheng, 1993);
$E_{pani}$ is the monthly water surface evaporation measured by pan evaporation, (mm); $H_i$ is the
mean annual groundwater depth, (m); $H_{max}$ is the critical groundwater depth at which the
phreatic evaporation will vanish (m); and $n$ is a dimensionless empirical coefficient related to
soil texture ranging from 1 to 3. Since the irrigation districts with groundwater depth less than
3 m only make up 1/5 of the total, the annual change of water storage were assumed to be
negligible, the sum of irrigation water, precipitation, and groundwater consumption can be used
as water availability for upper soil layers based on water balance in large irrigation districts
(refer to Fig.3B in study of Chen et al. (2020)), similar to the equivalent precipitation $P_e$
defined by Wang (2012):

$$P_e = I + P + ET_{gw} \tag{4}$$

where $ET_{gw}$ is groundwater evaporation, indicating the contribution of shallow
groundwater to $ET$. When the groundwater depth is larger than 3 m, no groundwater
contributes to evaporate any more, i.e., $ET_{gw} = 0$. With the newly defined $P_e$, the Fu's equation
can be expressed as:





$$\frac{ET}{P_e} = 1 + \frac{ET_0}{P_e} - \left[1 + \left(\frac{ET_0}{P_e}\right)^\omega\right]^{1/\omega} \qquad (5)$$

Affected by the natural factors, the variation of $ET$ can be determined by the variation in

water availability and energy supply while the land surface conditions for given districts are

assumed to be constant (Yang et al., 2006). The relative magnitude of $\frac{\partial ET}{\partial P_e}$ and $\frac{\partial ET}{\partial ET_0}$ can reflect

the relative effect of $P_e$ and $ET_0$ on $ET$ variation, respectively (Han et al., 2011):

$$\delta ET = \frac{\partial ET}{\partial P_e} \delta P_e + \frac{\partial ET}{\partial ET_0} \delta ET_0 \qquad (6a)$$

$$\frac{\partial ET}{\partial P_e} = 1 - \left[1 + \left(\frac{P_e}{ET_0}\right)^\omega\right]^{\frac{1}{\omega}-1} \left(\frac{P_e}{ET_0}\right)^{\omega-1} \qquad (6b)$$

$$\frac{\partial ET}{\partial ET_0} = 1 - \left[1 + \left(\frac{ET_0}{P_e}\right)^\omega\right]^{\frac{1}{\omega}-1} \left(\frac{ET_0}{P_e}\right)^{\omega-1} \qquad (6c)$$

where $\delta ET$, $\delta P_e$ and $\delta ET_0$ are the variability in actual evapotranspiration, water availability

and energy supply. The elasticity, defined as the indicator to reflect the sensitivity of

dependent variable to the change in other variables, is further applied to separate and evaluate

the influence of irrigation water, precipitation, and energy supply on the variation of $ET$:

$$S_I = \frac{\partial ET}{\partial I} \frac{I}{ET} \qquad (7a)$$

$$S_P = \frac{\partial ET}{\partial P} \frac{P}{ET} \qquad (7b)$$

$$S_{ET0} = \frac{\partial ET}{\partial ET_0} \frac{ET_0}{ET} \qquad (7c)$$

where $S_I$, $S_P$, and $S_{ET0}$ are the elasticities of evapotranspiration to irrigation water,

precipitation and energy supply, respectively. According to the definition, a positive elasticity

indicates that an increase in the independent variables will bring an increase in $ET$. The impact

of groundwater variation on $ET$ is not discussed in this study as the groundwater evaporation

in most of irrigation districts were out of consideration.

**2.3 Methodology for estimating actual evapotranspiration**





According to the report released by China Irrigation and Drainage Development Centre
(Dang and Feng, 2016), the measured net irrigation water denotes the fraction of total irrigation
water that stored in soil layers and available for crop growth, which is divided by the gross
irrigation water to obtain the irrigation water use efficiency (Dang and Feng, 2016). The
remaining serves to leach accumulated salt from soil surface, leaks or evaporates
unproductively. Similarly, the fraction of precipitation actually used by plants or
evapotranspiration can be approximately estimated following the U.S. Department of
Agriculture Soil Conservation Method (Smith, 1992), as widely used by numerous study and
crop models including in China (Cao et al., 2014a; Cao et al., 2014b):
$$P_{effd} = P_d(4.17 - 0.2P_d)/4.17 \qquad \text{for } P_d < 8.3 \text{ mm/d}$$
$$P_{effd} = 4.17 + 0.1P_d \qquad \text{for } P_d \geqslant 8.3 \text{ mm/d} \qquad (8)$$
where $P_{effd}$ is the daily effective precipitation, mm/d; $P_d$ is the actual daily precipitation,
mm/d. The sum of daily values is regarded as annual values. Assuming the variation of soil
water storage is negligible, the water balance equation at annual scale for irrigation district is
expressed as follow:
$$ET = I + P - D_i \qquad (9)$$
where $D_i$ is the outflow of irrigation districts that cannot be used as crop water consumption,
including deep seepage, runoff and drainage through ditches. The actual evapotranspiration
can be further approximated as the sum of net irrigation water and effective precipitation (Döll
and Siebert, 2002; Smith, 1992):
$$ET = I_{net} + \sum P_{effd} = \eta \times I + \sum P_{effd} \qquad (10)$$
where $I_{net}$ is net irrigation water, mm; $\sum P_{effd}$ is the annual effective precipitation calculated
as the sum of daily values, mm.





## 3. Results

### 3.1 Validation of water balance equation

The values of annual *ET* derived from MOD16 products from the year of 2010-2017 are used to validate the accuracy of water balance equation. As shown in Fig.2, the water balance equation performed well in estimating the values of *ET* compared with that of MOD16 products with RMSE of 124.4 mm, MRE of 18.6%, and $R^2$ of 0.6. It's reasonable to believe that the simulated results of water balance equations were accurate in the following study.

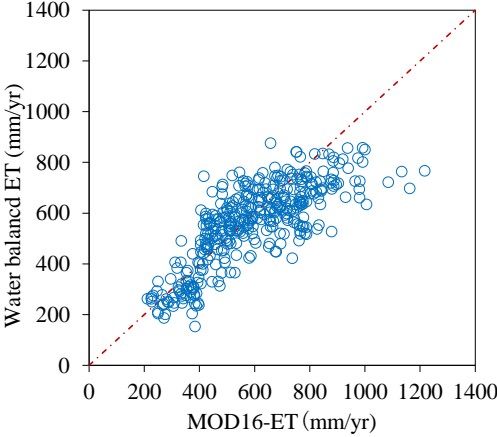

Fig.2 Comparison of annual *ET* between water balance equation and MOD16 products

### 3.2 Analysis of annual Budyko curves

With the use of equivalent precipitation, the ratio of evapotranspiration to the water availability for the irrigation districts is plotted against the ratio of potential evapotranspiration to the water availability as shown in Fig.3. The discrete data were observed in arid and semi-arid regions contrasted to relatively convergent data in the humid and semi-humid regions. The ranges of ω values derived from the data scatted in arid and semi-arid areas varied from 1.25 to 2 and 1.4 to 2 respectively, while those in humid and semi-humid areas were relatively stable. The distinguishing performances of data in various climatic regions are mainly attributed to




the different dominant roles on evapotranspiration under various climatic conditions. The
dominant role of energy supply in evapotranspiration variation in humid and semi-humid
regions was highlighted via the form of $ET/P_e \sim ET_0/P_e$ since $ET_0$ was placed in the position
of numerator, leading to the convergently distributed data trend with unfluctuating values of ω.
Similarly, the control role of water availability on $ET$ in arid and semi-arid regions was
weakened by the exaggerated influence of catchment characteristics through the form of
$ET/P_e \sim ET_0/P_e$ as $P_e$ was placed in the position of denominator, mainly reflected in the
different values of ω and dispersion of data points (Yang et al., 2007).

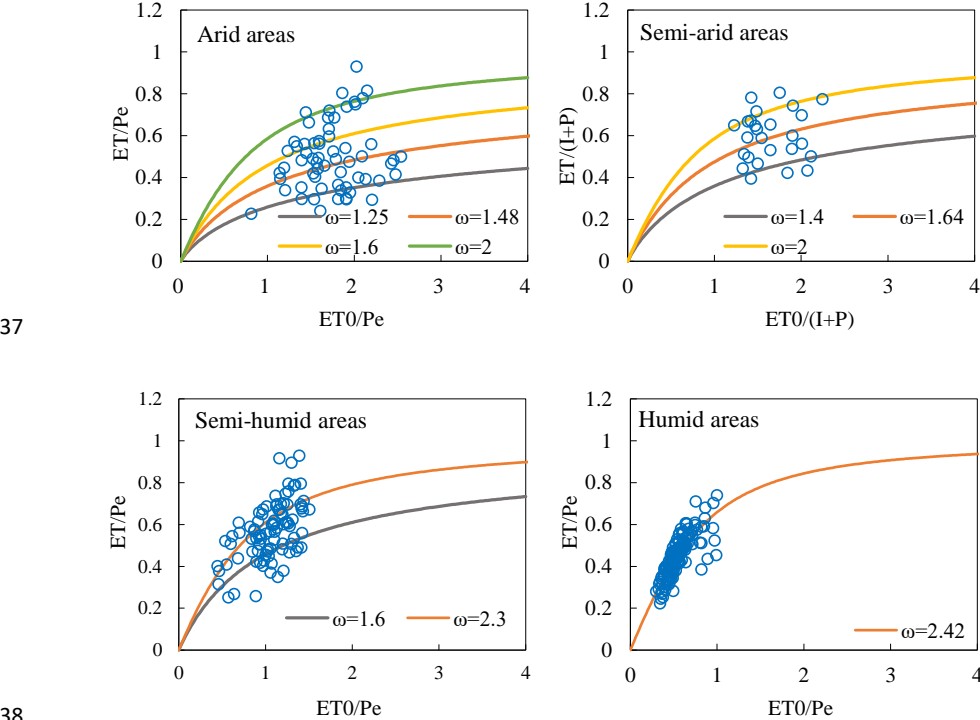



Fig.3 Mean annual values of actual evapotranspiration, potential evapotranspiration, and
equivalent precipitation data plotted in Fu's equation over 2010-2017 for irrigation areas
under various climate conditions
**3.2 Controlling factors on the variation of *ET***


Fig.4 shows the relationship between $\partial ET/\partial ET_0$ and $ET_0/P_e$, as well as the relationship
between $\partial ET/\partial P_e$ and $ET_0/P_e$ under various climatic conditions, respectively. All the
irrigation districts are further classified into three climate conditions as follows: water-limited
condition ($ET_0/P_e > 1.35$), equitant condition ($0.76 < ET_0/P_e \leqslant 1.35$), and energy-limited
condition ($ET_0/P_e \leqslant 0.76$) (McVicar et al., 2012). Under the water-limited condition, the values
of $\partial ET/\partial ET_0$ are smaller than $\partial ET/\partial P_e$ and insensitive to the variation of ω, highlighting the
dominant role of water availability on evapotranspiration; under energy-limited condition, the
values of $\partial ET/\partial P_e$ are smaller than $\partial ET/\partial ET_0$ and insensitive to the variation of ω,
highlighting the dominant role of energy supply; under equitant condition, the overlaps of
plotted points are observed and emphasize the combined effect of water availability and energy
supply on $ET$ variation. These results are consistent with original Budyko hypothesis
(Carmona et al., 2016; Fu, 1981; Zhang et al., 2001; Zhang et al., 2004).

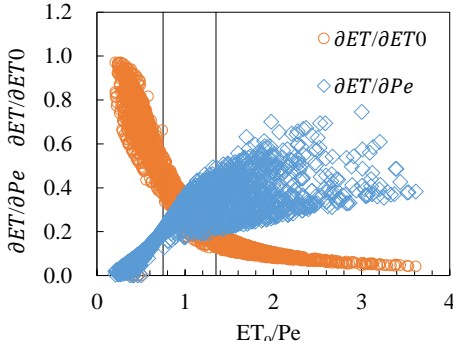


Fig.4 Plot of $\frac{\partial ET}{\partial P_e}$ and $\frac{\partial ET}{\partial ET_0}$ with aridity index ($\frac{ET_0}{P_e}$) for the large irrigation districts in China
**3.3 Sensitivity of $ET$ to $I$, $P$ and $ET_0$**
Fig.5 shows the variation of elasticities among 371 irrigation districts and the statistical
results are grouped by arid, semi-arid, semi-humid and humid conditions. The larger values of
$S_{ET_0}$ and $S_{I+P}$ occurred respectively in non-arid (humid and semi-humid) areas and non-humid



(arid and semi-arid) areas, suggesting that the variation of $ET$ is more sensitive to energy
supply in humid and semi-humid areas or water availability in arid and semi-arid areas. Except
in arid areas, the values of $S_I$ is generally smaller than $S_P$. This is because the irrigation water
serves as the main water resource for crop growth in arid areas due to the severe water shortage
(the mean annual precipitation is 95.6 mm/yr) but supplemental water of precipitation in other
climatic conditions. The mean values of $S_I$ for four climatic conditions are 0.529, 0.350, 0.118,
and 0.038; and the mean values of $S_P$ are 0.097, 0.290, 0.216, and 0.111. These results indicate
that a 10% increase in irrigation water could cause evapotranspiration increase by 5.29%,
3.50%, 1.18%, and 0.38% in arid, semi-arid, semi-humid and humid areas; a 10% increase in
precipitation could cause evapotranspiration increase by 0.97%, 2.90% 2.16%, and 1.11%.
Similarly, a 10% increase in potential evaporation could cause evapotranspiration increase by
3.93%, 3.83%, 6.79%, and 8.61%, respectively.



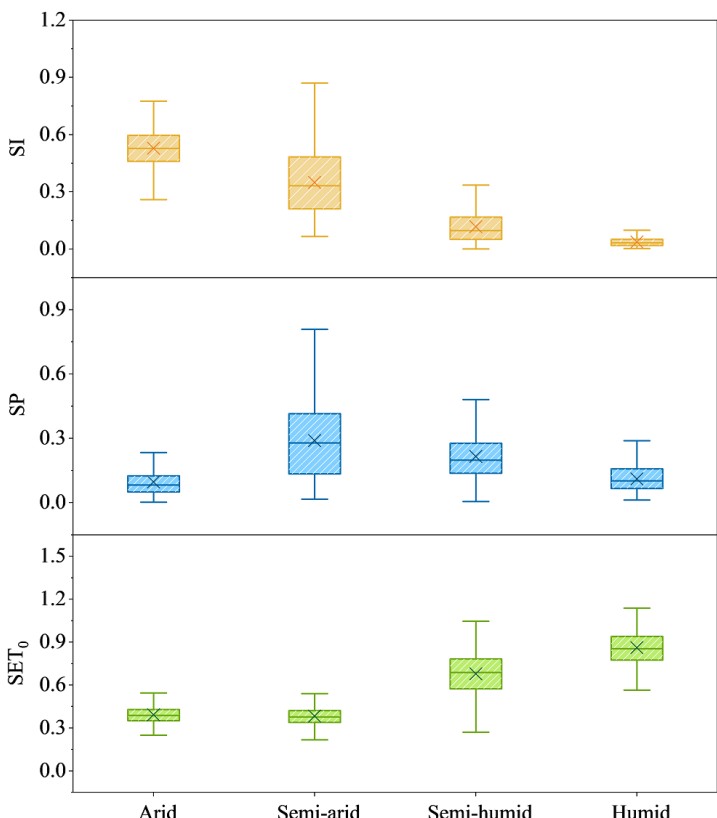


Fig.5 Comparison between elasticities of irrigation water ($S_I$), precipitation ($S_P$) and energy
supply ($S_{ET0}$) for arid, semi-arid, semi-humid and humid regions in China
**3.4 Characteristics of ω and influence factors**
As a parameter to represent the integrated effects of catchment characteristics on *ET*
variation, the optimal values of ω for all irrigation districts were obtained by minimizing the
values of RMSE between the Budyko modelled annual $ET/ET_0$ and the estimated ones using
the data from 2010-2017. The values of ω in humid and semi-humid regions are generally
larger than those in arid and semi-arid regions (Fig.6). Range of the ω values obtained from
this study (1.24-3.34) compares favourably with the results from previous studies. Applying a
neural network model in 224 small basins (ranging from ~100 to 10000 km²) with the data



from MOPEX, Xu et al. (2013) observed that the values of ω varied from 1.0 to 4.9 with a
median value of 2.6. Based on the features of vegetation cover, Zhang et al. (2004) selected
forested and grassed catchments as two typical types and found the averaged ω values for the
two catchments were 2.84 and 2.55 with data ranging between 1.7 to 5.0. In China, the range
of ω obtained from 108 arid and semi-arid basins varied from 1.3 to 4.6 (Yang et al., 2007), a
little higher than the results from 97 basins in Australia (1.8-3.8) using Murray Darling Basin
data (Donohue et al., 2011).

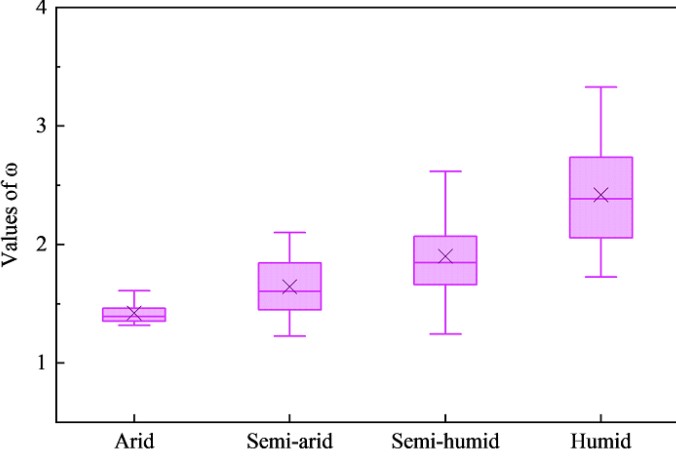


Fig.6 Distribution of the optimal ω values grouped by four climate conditions

Besides the water availability and energy supply, the catchments characteristics also play

a significant role in determining the shape of Budyko curves (Potter and Zhang, 2009; Roderick
and Farquhar, 2011; Woods, 2003; Yuan et al., 2010). Since more than 95% of irrigation
districts locate in plain areas with slopes ranging from 1 to 5 degree, the influence of terrestrial
slope on ω is neglected in this study. The mean values of NDVI were selected to represent the
differences in vegetation cover across the irrigation districts. The vegetation coverage in
southeast of China is significantly higher than that in northwest with mean NDVI values of
0.51 and 0.17, respectively (Fig.1B). Meanwhile, the proportion of clay and sand is supposed





to be another influence factor on *ET* variation owing to the different water-holding capacities
(Fig.1CD). For dimensional analysis, the ratio of clay to sand content expressed as $P_{cl}/P_{sa}$ is
used to represent the soil property. As shown in Fig.7, the parameter ω in Fu's equation is
closely correlated with the long-term vegetation coverage and soil property. Using the multiple
linear regression analysis method (MLRA), the empirical equation of parameter ω can be
determined as follows:
$$\omega = 0.537 exp\,(2.08NDVI)(\frac{P_{clay}}{P_{sand}})^{0.28} + 1 \qquad (11)$$
As shown in Fig.8A, the model explains 48% of the observed variance with R of 0.693 and
RMSE of 0.315.

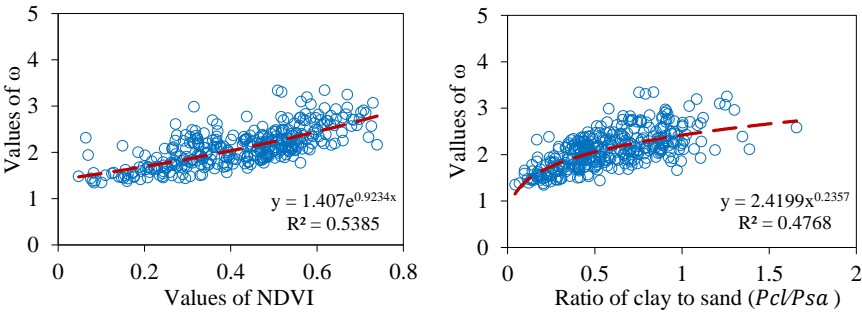

Fig.7 Relationships between ω and NDVI and soil texture
Various models have been proposed to establish relationship between ω and influence
factors, including soil hydrological features, terrestrial slope, vegetation coverage and climatic
factors (Donohue et al., 2012; Li et al., 2013; Shao et al., 2012; Yang et al., 2009; Yang et al.,
2007). However, in the view of the consideration of climatic conditions in the original Budyko
framework, it's better to estimate ω with climate factors excluded to avoid cross-correlation
issues (Xu et al., 2013). Also, the expression of soil texture denoted by the ratio of clay and
sand content is more accessible than other soil properties such as relative infiltration capacity
and relative soil water storage (Yang et al., 2009). With the estimated values of parameter ω,




Fu's equation reproduced mean annual $ET$ well for irrigation districts with $R^2$ of 0.74 and
RMSE of 63.08 mm/yr (Fig.8B).

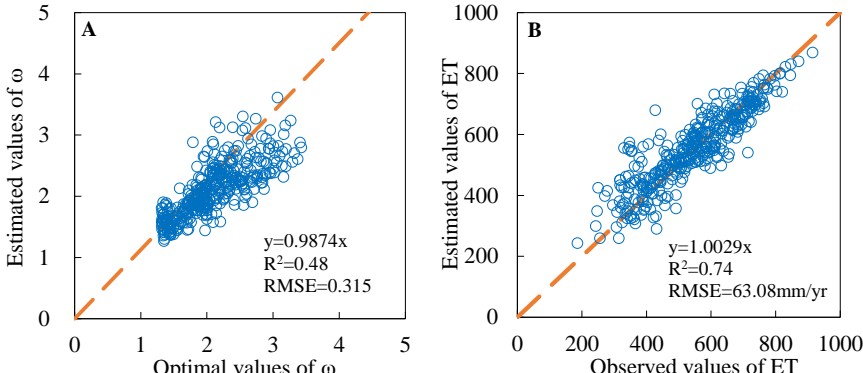


Fig.8 A: Comparison between optimal values of ω and estimated ones from multiple linear
regression analysis; B: Relationship between actual $ET$ and estimated $ET$ using ω calculatd by
Equation (11)
**4. Discussion**
**4.1 Uncertainty about the annual water balance and influence factors on ω**

For long-term water balance in large natural catchments, evapotranspiration can be

regarded as the partitioning of precipitation which is serving as water availability in Budyko
formulations ($ET = P - R$) while water storage is assumed to be negligible (Donohue et al.,
2010; Hobbins et al., 2001; Rodell, 2004; Xue et al., 2013). Recently, the estimation of water
balance at finer time scales has attracted more attentions in many studies and these studies
showed that the water storage change (including soil moisture and groundwater) played a
significant role in annual water balance and made a great contribution to meet the deficit of
water supply for crop water demand (Chen et al., 2018; Flerchinger and Cooley, 2000;
Ghamarnia et al., 2013; Leblanc et al., 2009; Namuburg et al., 2005; Valayamkunnath et al.,
2019). The application of equivalent precipitation incorporating water storage change is able

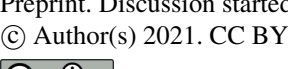



to work better at improving the performance of Budyko predictions in annual scale (Chen et
al., 2013; Istanbulluoglu et al., 2012; Wang, 2012; Wang and Zhou, 2016), especially for basins
in arid and semi-arid regions (Du et al., 2016; Milly and Dunne, 2002; Xing et al., 2018). In
this study, we assume the water storage changes at annual scale are negligible, which may lead
to errors in accurate estimation of water availability, and then influence the analysis about the
shape of Budyko curves and the variation of ω values. Furthermore, the uncertainty of the
influence factors on the variation of ω should be recognized as well. The variation of ω in
irrigation districts is associated with NDVI and soil property. Due to the differences in water
requirement between crop types, however, the evapotranspiration from the pixels with same
NDVI differs as a response to crop planting patterns (Bai et al., 2017; Eichelmann et al., 2018;
Mo et al., 2015). Thus, the influence of water storage change and crop planting patterns (i.e.,
the fraction of each crop type) on the allocation of water availability should be accommodated
for detailed analysis in further study.
**4.2 Water use efficiency in irrigation districts**

By selecting 108 arid and semi-arid catchments and 102 humid catchments in China,

Wang et al. (2018) estimated that the mean values of ω for normal and karst humid catchments
are 2.23 and 2.03, while for arid and semi-arid catchments is 3.18. In present study, however,
the results shown in Fig.6 indicate that the values of ω increased from arid to humid regions.
The smaller values of ω in arid and semi-areas produced smaller values of $ET/P_e$ varying from
0.4 to 0.6. In this context, $ET$ is hardly approaching to the total water availability even under
extremely arid condition. As shown in Fig.9, only 30% to 70% of rainfall are consumed for
crop use and the effective precipitation efficiency in arid and semi-arid regions are generally
larger than those in humid and semi-humid regions. The arid and semi-arid regions are
generally suffering from severe soil salinization (Jiang and Shu, 2018; Peng et al., 2019; Qian
et al., 2019) and a series of ecological environment problems caused by it (Besser et al., 2017;



Haj-Amor et al., 2017) owing to the scarcity of rainfall and high potential evaporation. To
alleviate the negative influence of soil salinization on crop yield, part of irrigation water is
applied to flush accumulated salt from soil surface to prepare for the next season's crop (Tang,
2018; Wei and Xu, 2005; Zhang, 1993), finally resulting in small fraction of water availability
used by $ET$. The amount of irrigation water used to leach salt mainly depends on local irrigation
technology and water management. In semi-humid and humid areas with relatively abundant
precipitation, the application of irrigation events aims to regulate the unevenly distributed
rainfall in a year. The small values of $ET/P_e$ reflect the generally low water use efficiency of
irrigation districts in China and indicate the significance of water saving measurements. For
arid and semi-arid areas with relatively higher rainwater utilization, the improvement of
drainage systems can effectively remove the accumulated salt from soil surface, further
reducing the fraction of irrigation water used to alleviate soil salinization. For semi-humid and
humid areas with enough precipitation, the judicious regulation of rainfall including the
effective rainwater harvesting and reuse is expected as helpful way to improve water use
efficiency. Reducing unproductive evapotranspiration through canal lining or drip irrigation is
applicable to improve water use efficiency as well.

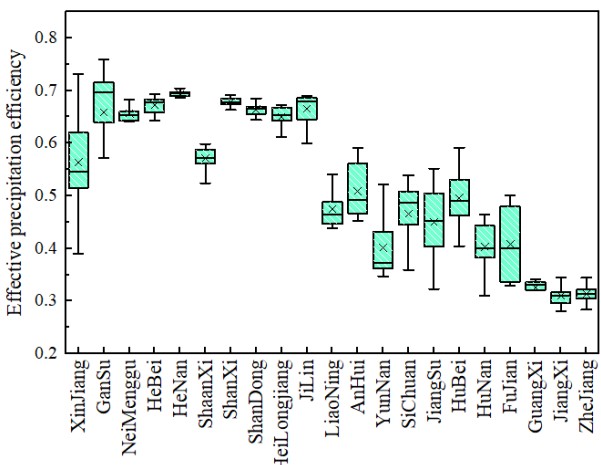

Fig.9 Range of effective precipitation efficiency across study area





## 5. Conclusions


This paper aims to examine the performance of Budyko framework in agricultural
irrigation regions using Fu's equation. A total of 371 large irrigation districts across China were
selected as study areas, which were grouped by arid, semi-arid, humid and semi-humid areas.
With the precipitation replaced by equivalent precipitation ($P \rightarrow P_e = (I + P + ET_{gw})$), the
data of $ET/P_e \sim ET_0/P_e$ were plotted well in Budyko areas. The values of ω increased from arid
and humid regions. Smaller values in arid and semi-arid regions were attributed to the low
water use efficiency. Part of irrigation water is not consumed by crop growth but serving to
leach salt to alleviate salinization. Surface runoff or deep seepage caused by concentrated
rainfall in humid and semi-humid regions is invalid for crop growth and irrigation is still
applied to regulate water deficit caused by it. Corresponding water-saving measures should be
taken to improve water use efficiency in different climatic areas. In this study, the variation of
ω was found to be closely related with NDVI and soil texture (denoted by the ratio of clay and
sand content, $P_{cl}/P_{sa}$). The simple empirical model of ω developed using NDVI and soli
properties performed well in reproducing $ET$ in irrigation districts.
**Appendix**
*Section 1 Estimation of irrigation water use efficiency η*
Irrigation water use efficiency $η$ denotes the fraction of the total irrigation water actually
used by crop growth, i.e., the ratio of net irrigation water to total irrigation water. Referring to
the report released by China Irrigation and Drainage Development Centre, two methods are
alternative in large irrigation districts to estimate the irrigation water use efficiency when
applied to at least three selected typical patches of each crop type: direct measurement method
and head-end measurement method (Fig.B1). For direct measurement, the increase in the depth
of water surface or in soil water content in soil moist layer (the applicability of methods
depends on crop types, i.e., dry farming or rice) after irrigation events is measured as the net
irrigation water; for head-end measurement, the difference between inflow to and outflow from
irrigation districts is regarded as the amount of net irrigation water. Details of the measurement
methods are shown in Tab.C1.

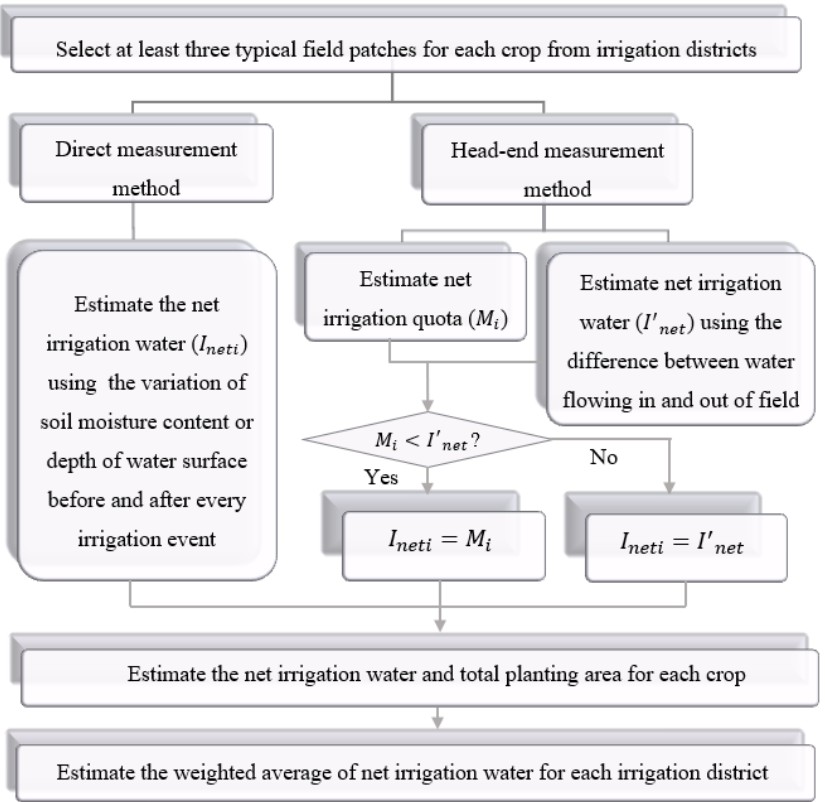

Fig.B1 Calculation process of net irrigation water for field patches and large irrigation
districts
Tab.C1 Methods for calculating irrigation water use efficiency for irrigation districts

| Crop type | | | Direct measurement method | Head-end measurement method |
|---|---|---|---|---|
| | Dry farming | | $I_{neti} = H(\theta_{v2} - \theta_{v1})$ | $I_{neti} = \min \left( \dfrac{I_{ini} - D_i}{A_i}, M_i \right)$ |
| | Rice | Submerged irrigation | $I_{neti1} = H_2 - H_1$ | |
| | | Damp irrigation | $I_{neti2} = H(\theta_{v2} - \theta_{v1})$ | |



| Net irrigation water of irrigation district: $I_{net} = \frac{\sum_i^n I_{neti} \cdot A_i}{\sum A_i}$ | Utilization coefficient: $\eta = \frac{I_{net}}{I}$ |

Note: $I_{neti}$ is the net irrigation water amount for each crop estimated from selected typical field
patches, mm; $i$ denotes the crop type; $I_{net}$ is the weighted average of net irrigation water for
the whole irrigation districts, mm; $I$ is the total irrigation water diverted from water-supplying
area for irrigation districts, mm. For dry farming and damp irrigation stage of rice fields, $H$ is
the depth of moist soil layer, mm; $\theta_{v1}$ and $\theta_{v2}$ are the soil volumetric water content before and
after irrigation events, %. For the submerged irrigation stage of rice fields, $H_1$ and $H_2$ are the
depth of water surface before and after irrigation events, mm. When applying the head-end
measurement method, the judgement of whether sufficient irrigation is the premise for accurate
estimation of net irrigation water. $I_{ini}$ and $D_i$ are the water amount flowing in and out of fields,
mm; $M_i$ is the estimated net irrigation quota for each crop, mm; and $A_i$ is the total cover area
of each crop. Notably, rice fields have no drainage during crop growth period.
*Data availability*
The observed data used in this study are not publicly accessible. These data have been collected
and supported by China Irrigation and Drainage Development Centre. Anyone who would like
to use these data should contact Hang Chen and Zailin Huo to obtain permission.
*Author contributions*
JC, YS and ZH provided the data. HC and ZH contributed to the development of the model.
Preparation and revision of the paper were done by ZH, under the supervision of LZ.
*Competing interests*
The authors declare that they have no conflict of interest.
*Acknowledgments*



We appreciate the contributions of the editor and anonymous reviewers whose comments and
suggestions significantly improve this article. Special thanks go to China Irrigation and
Drainage Development Centre for their support and providing information and data.
*Financial support*
This research has been supported by the National Key Research and Development Program of
China (2018YFC0407703; 2016YFC0400107) and the National Natural Science Foundation
of China (51639009; 51679236).

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
