# Peer review of "The application of Budyko framework to irrigation districts in China"

_Hydrology and Earth System Sciences, 2021_

## Author Comment (AC1)

**Revision Notes (HESS-2021-80)**

**Responses to the comments of Reviewer #1:**

We would like to thank reviewer 1 for his extensive and thoughtful comments. In this document we give a detailed response to all comments. Below we cite first the comment, this is followed by our response and often by a section how the text will be revised in the manuscript. The text in blue are changes and additions in the original text. For clarity we do not show any of the removed text.

Thank you so much.
Zailin Huo

General Comments:

This article presents a study that extends the Budyko framework to irrigated areas and applies it on several districts in China. The topic is clearly relevant for HESS and this article may guide further studies aiming at taking into account irrigated areas in hydrological modeling studies. But at this stage, the proposed study relies on many hypotheses that are not tested/mentioned clearly and consequently, the reach of the results is difficult to assess.

1. Applicability of the Budyko hypothesis in unclosed systems

The Budyko framework is usually intended to describe/model the partitioning of water fluxes at the catchment scale. The catchment scale is important since it allows to work on a closed system, where inputs and outputs can be clearly and unequivocally stated. In the proposed study, the Budyko framework is applied on irrigation districts and the different fluxes considered are, to my opinion inter-dependent. For example, in Eq. 4, equivalent precipitation is proposed as the sum of Precipitation, Irrigation and groundwater consumption. This suggests implicitly that the Irrigation water and Groundwater used for evapotranspiration are water fluxes originated from other sources than Precipitation falling over the district area, which is questionable. Consequently, using Eq.4 may lead to count Precipitation fluxes twice since water provided by irrigation (and possibly groundwater) originate from precipitation. This may be true depending on spatial and temporal scales considered but this is not discussed in the paper. Both references cited in the paper to present Eq. 4 are not relevant since Wang et al. (2011) did not consider irrigation but water storage and Chen et al. (2020) considered catchment scale modelling.

**Response:** Thanks for your suggestions. In our study, the irrigation data were measured and provided by China Irrigation and Drainage Development Center. Based on the issued report, the total amount of water diverted from external water resource was measured as irrigation water, indicating that the irrigation water was reasonable to be

regarded as water input independent on precipitation. We agree with you the use of equation 4 is likely to cause double count of precipitation since the groundwater may be recharged by precipitation. Here, we will explain the independent role of groundwater uptake in water balance based on the new perspective to conduct water balance for vadose zone proposed only in our previous study (Chen et al., 2020). "The groundwater uptake is defined as the upward movement of groundwater induced by active crop growth and strong evaporation. While the aquifer was divided into soil layer and groundwater aquifer, the exchange flow between them should be considered as components for water balance analysis" (Fig.B2 in Appendix in revised manuscript, cited from Fig.3 in our previous study (Chen et al., 2020)). The groundwater recharge (GR) from infiltration of soil layer is supposed to be output component of water balance (Fig.B2 A). "For areas with groundwater depth less than 3 m, the upward movement of groundwater served as input component for water balance together with precipitation and irrigation water (Fig.B2 B). Evapotranspiration and infiltration to groundwater were taken as output components of water balance, together leading to the variation in soil water storage (Fig.B2 B)". In our previous study (Chen et al., 2020), it has been proven that 80% of total annual groundwater uptake with shallow groundwater occurred during April to September while the groundwater levels presented a continuously declining trend. According to the long term observation data of groundwater level, the visible rises in groundwater levels after each irrigation and precipitation event ranged from 0.01 to 0.1 m, indicating that most of irrigation water and precipitation tend to evaporate before infiltrating to groundwater during crop growing period due to the heavy crop water demand and strong evaporation (Chen et al., 2020). "Furthermore, the use of averaged groundwater depth offsets the effect of potential errors to an extent". Based on the above conclusions, "it is acceptable to neglect the double counting of water and regard groundwater uptake as a new component in water availability independent on irrigation and precipitation" in this study. In addition, the groundwater depths in 4/5 of the total irrigation districts were more than 3 m, where the groundwater consumption was negligible while only irrigation water and precipitation were considered as water availability. Considering the distribution of irrigation districts across China in this study, the accuracy of the water balance meets the precision requirements at national scale. In this study, the citation of Wang et al. (2011) tended to introduce the definition of equivalent precipitation. We have cancelled the citation of Wang et al. (2011) in revised manuscript to avoid ambiguity.

[Figure]

Fig.B2 Structure of water balance model in areas with deep groundwater (A) and shallow groundwater (B) (cited from the Fig.3 in study of Chen et al. (2020))

2. Lack of clear validation with observed data

The authors propose a validation of ET using MOD16 product but it should be stated that the comparison to MOD16 ET cannot be viewed as a strict validation since MOD16 ET relies heavily on modelling. The validation using streamflow time series at catchment scales is to my opinion the unique way to perform a real validation with independent data. The interpretation of Fig.3 is also complicated since all variables (ET/Pe, ET0/Pe) are derived from computation with associated uncertainties that are very difficult to quantify at this stage. Interpreting the deviation of the simulations and "observations" is thus impossible.

**Response:** Thanks for your suggestions. As far as possible, we tend to understand the variation pattern of evapotranspiration (ET) in large irrigation districts and conduct attribution analysis at national scale. Actually, it is extremely difficult to calculate actual evapotranspiration for the national scale. Here we use the water balance method with measured irrigation water to calculate ET, which we think is relatively accurate for the national scale under current conditions. We agree with you that "the comparison to MOD16 ET cannot be viewed as a strict validation", so we changed the "Validation of water balance equation" into "Comparison between water balance ET and MOD16 ET" as the subtitle of section 3.1 in revised manuscript. The relative expression about "validation" has been rephrased in revised manuscript as well. In the revised manuscript, it was revised as:

"3.1 Comparison between water balance ET and MOD16 ET

The values of annual ET derived from MOD16 product during the year of 2010-2017 are used to make comparison with those estimated by water balance method (Equation 10). As shown in Fig.2, the water balance equation performed well in estimating ET under current conditions compared with that of MOD16 product with RMSE of 124.4 mm/yr, MRE of 18.6%, and R2 of 0.6. In terms of national scale, it's reasonable to believe that the estimated

results of water balance equation were relatively accurate in the following study."

Other comments

1. L52-53: Phrasing problems.

**Response:** Sorry for the phrasing problems. We have revised the sentence as "many studies subsequently derived parametric Budyko-type formulations" in revised manuscript.

2. L151: Why considering Pan evaporation in Eq.3 instead of Penman equation?

**Response:** Sorry for the ambiguous statement about phreatic evaporation equation. In the original Aver'yanov's phreatic equation, the water surface evaporation rather than potential evaporation was used to estimate groundwater evaporation. Thus, the pan evaporation (which is generally used to denote water surface evaporation) is considered instead of potential evaporation estimated by Penman equation. The explanation is revised as "Epani is the monthly water surface evaporation measured by pan evaporation, (mm)" in revised manuscript.

3. L197-198: Computing effective rainfall is highly uncertain. Eq. 8 is a way to estimate it but may leads to large errors. The USDA SCS method provides alternative ways to take into account soil types and land use classes. Besides, I failed to understand why ET0 is not involved in this calculus of effective rainfall. I would expect that the authors quantify the uncertainties related to this estimation, or at least provide the magnitude of effective rainfall compared to rainfall and Irrigated fluxes

**Response:** Thanks for your suggestion. According to the definition proposed by the Soil Conservation Service of U.S.D.A. (1967), effective rainfall is that which is received during the growing period of a crop and is available to meet consumptive water requirements. It does not include surface run-off or deep percolation losses (Dastane, 1978). The U.S. Department of Agriculture's Soil Conservation Service has developed a procedure for estimating effective rainfall by processing long term climatic and soil moisture data. In the absence of detailed data, however, "the empirical equation proposed by U.S. Department of Agriculture's Soil Conservation Service was suggested to estimate effective precipitation in areas with slope less than 5°, which has been widely used and performed well in numerous studies and crop models including in China (Cao et al., 2014a; Cao et al., 2014b; Qin et al., 2016; Zheng et al., 2020)".

In this study, the magnitude of effective precipitation compared to precipitation was expressed as the ratio of effective precipitation to the total annual precipitation, which was defined as effective precipitation efficiency as shown in Fig.9. "Only 30% to 70% of rainfall are consumed for crop use and the effective precipitation efficiency in arid and semi-arid regions are generally larger than those in humid and semi-humid regions". In the original Budyko formula, the evapotranspiration ratio ($ET/P$) denotes the partition of precipitation used for evapotranspiration, i.e., the precipitation use

efficiency for plant growth and soil evaporation under natural condition. In our study, the actual evapotranspiration is estimated as the sum of net irrigation water ($I_{net}$) and effective precipitation ($P_{eff}$) shown in equation 10. The expression of $ET/P$ can be transferred into $(I_{net} + P_{eff})/P_e$. Since the values of ω are determined by the relationship between $ET/P_e$ and $ET_0/P_e$, we try to attribute the relatively smaller values of ω obtained for irrigation districts to the water use efficiency, including the irrigation water use efficiency and effective precipitation efficiency. The related analysis results are shown in Section 4.2 "Water use efficiency in irrigation districts" in revised manuscript.

4. L219: Typo in y-axis label.

**Response:** Sorry for the typo in y-axis label. We have revised the y-axis label as "Water balance ET (mm/yr)" in revised manuscript.

5. L237: Perhaps I missed something but why Pe is replaced with (I+P) in Semi-arid areas?

**Response:** Sorry for the typo. The water availability was used as Pe, i.e., the sum of irrigation water, precipitation, and groundwater consumption. Only for the areas with groundwater depth more than 3 m, Pe was replaced with (I+P) since the groundwater evaporation was negligible. We have revised $ET_0$/(I+P) for semi-arid areas as $ET_0$/Pe in Fig.3 in revised manuscript.

6. L279-281. Is there a clear justification why w is different according to the climatic settings? I would expect that w be more likely dependent on land use, soil and vegetation types, not climate.

**Response:** Thanks for your suggestions. We agree with you that the parameter ω were dependent on land use, soil, and vegetation types. According to the study of Wang et al. (2018)., it has been proven that the values of parameter ω varied greatly among different climatic conditions. In our study, the areas were firstly classified into four climatic conditions according to the values of aridity index, including arid area, semi-arid area, humid area, and semi-humid area. Meanwhile, the estimated values of parameter ω were grouped by climatic conditions as well to further explore the relationship between parameter ω and influence factors related to the climatic conditions. Since the study areas were irrigated districts and 95% of them were located in plain area with slope less than 5°, NDVI and soil texture were selected as influence factors. The relevant analysis results about the influence of land surface characteristics on parameter ω were discussed in Section 3.4 "Characteristics of ω and influence factors" in revised manuscript.

7. L379: "Effective precipitation efficiency" is not clearly defined. How is it computed and what is really shown on Fig.9?

**Response:** Sorry for the missing information about "effective precipitation efficiency". "Similar to the definition of irrigation water use efficiency, the effective precipitation efficiency is defined as the ratio of effective precipitation to the total annual

precipitation ($P_{eff}/P$)". In the original Budyko formula, the evapotranspiration ratio ($ET/P$) denotes the partition of precipitation used for evapotranspiration, i.e., the precipitation use efficiency for plant growth and soil evaporation under natural condition. In our study, the actual evapotranspiration is estimated as the sum of net irrigation water ($I_{net}$) and effective precipitation ($P_{eff}$) shown in equation 10. The expression of $ET/P$ can be transferred into $(I_{net} + P_{eff})/P_e$. Since the values of ω are determined by the relationship between $ET/P_e$ and $ET_0/P_e$, we try to attribute the relatively smaller values of ω obtained for irrigation districts to the water use efficiency, including the irrigation water use efficiency and effective precipitation efficiency. The explanation about Fig.9 is revised as follow:

"As shown in Fig.9, only 30% to 70% of rainfall are consumed for crop use and the effective precipitation efficiency in arid and semi-arid regions are generally larger than those in humid and semi-humid regions. The arid and semi-arid regions are generally suffering from severe soil salinization (Jiang and Shu, 2018; Peng et al., 2019; Qian et al., 2019) and a series of ecological environment problems caused by it (Besser et al., 2017; Haj-Amor et al., 2017) owing to the scarcity of rainfall and high potential evaporation. To alleviate the negative influence of soil salinization on crop yield, part of irrigation water is applied to flush accumulated salt from soil surface to prepare for the next season's crop (Tang, 2018; Wei and Xu, 2005; Zhang, 1993), finally resulting in small fraction of water availability used by ET. The amount of irrigation water used to leach salt mainly depends on local irrigation technology and water management. In semi-humid and humid areas with relatively abundant precipitation, the application of irrigation events aims to regulate the unevenly distributed rainfall in a year. The small values of ET/Pe reflect the generally low water use efficiency of irrigation districts in China and indicate the significance of water saving measurements. For arid and semi-arid areas with relatively higher rainwater utilization, the improvement of drainage systems can effectively remove the accumulated salt from soil surface, further reducing the fraction of irrigation water used to alleviate soil salinization. For semi-humid and humid areas with enough precipitation, the judicious regulation of rainfall including the effective rainwater harvesting and reuse is expected as helpful way to improve water use efficiency. Reducing unproductive evapotranspiration through canal lining or drip irrigation is applicable to improve water use efficiency as well."

**Reference:**

Cao, X.C., Wu, P.T., Wang, Y.B. and Zhao, X.N., 2014a. Assessing blue and green water utilisation in wheat production of China from the perspectives of water footprint and total water use. Hydrology and Earth System Sciences, 18(8): 3165-3178.

Cao, X.C., Wu, P.T., Wang, Y.B. and Zhao, X.N., 2014b. Water footprint of grain product in irrigated farmland of China. Water Resources Management, 28(8):

2213-2227.

Chen, H., Huo, Z.L., Zhang, L. and White, I., 2020. New perspective about application of extended Budyko formula in arid irrigation district with shallow groundwater. Journal of Hydrology, 582.

Dastane, N.G., 1978. Effective rainfall in irrigated agriculture. Food and Agriculture Organization of the United Nations, Rome, Italy.

Qin, L., Jin, Y., Duan, P. and He, H., 2016. Field-based experimental water footprint study of sunflower growth in a semi-arid region of China. J Sci Food Agric, 96(9): 3266-73.

Wang, T.T. et al., 2018. The predictability of annual evapotranspiration and runoff in humid and nonhumid catchments over China: comparison and quantification. Journal of Hydrometeorology, 19(3): 533-545.

Zheng, X.X., Qin, L.J. and He, H.S., 2020. Impacts of Climatic and Agricultural Input Factors on the Water Footprint of Crop Production in Jilin Province, China. Sustainability, 12(17).

---

## Author Comment (AC2)

**Revision Notes (HESS-2021-80)**

**Responses to the comments of Reviewer #2:**

We would like to thank reviewer 2 for his extensive and thoughtful comments. In this document we give a detailed response to all comments. Below we cite first the comment, this is followed by our response and often by a section how the text will be revised in the manuscript. The text in blue are changes and additions in the original text. For clarity we do not show any of the removed text.

Thank you so much.
Zailin Huo

General comments:

1. The authors present a study where they investigate the role of irrigation in the Budyko framework. This is a long-standing issue in Budyko framework that is not solved yet (see e.g., Han et al 2011; Mianabaid et al 2020), mainly due to the lack of data on irrigation. This paper is fortunate to have irrigation data for 371 catchments in China, and is therefore unique in its kind. However, this data is not public as stated in the 'data availability' section. In my opinion the given author statement is not in line with HESS's data policy (https://www.hydrology-and-earth-system-sciences.net/policies/data_policy.html). Additionally, irrigation data is also more complex in comparison to hydrometeorological data, as the data highly depends on irrigation type, associated losses, etc. I appreciate the authors effort to explain the underlying principles in the Appendix; however, this information is not enough (e.g., how is the water use efficiency calculated or measured??; is the irrigation water originating from the same catchment?). This results in the fact that I cannot judge the validity of the data (which is essentially modelled irrigation), which is at the core of this study. Hence, I highly recommend to share the data so that your study can be verified.

Mianabadi A., Davary, K., Pourreza-Bilondi, M., Coenders-Gerrits, A. M. J., 2020. Budyko framework; towards non-steady state conditions. Journal of Hydrology. https://doi.org/10.1016/j.jhydrol.2020.125089

Han, S., Hu, H., Yang, D., & Liu, Q. (2011). Irrigation impact on annual water balance of the oases in tarim basin, northwest china. Hydrological Processes, 25(2), 167-174. doi:10.1002/hyp.7830

**Response:** Thanks for your suggestions and sorry for the ambiguous description about the irrigation data. The annual data of total irrigation water and irrigation water use efficiency "were measured and provided by China Irrigation and Drainage Development". Annual irrigation water use efficiency is calculated as the ratio of net irrigation water to total irrigation water. According to the report issued by China Irrigation and Drainage Development Centre, "the net irrigation water referred to the net water amount used by crop growth and soil evaporation". As to the measurement methods, "at least three typical field patches for each crop type in every large irrigation district with clear boundary, regular shape and moderate area were selected to calculate net irrigation water, while the crop types, irrigation types, soil texture, irrigation methods, and groundwater depth were taken into account comprehensively. The field patches were selected in the control range of delivery canals along upper, middle, and lower reaches, respectively". The detailed measurement methods for net irrigation water are shown in following table (i.e., Tab.C1 in Appendix in revised manuscript). For various crop types (dry farming or rice), the difference of soil water content in moist soil layer or water surface depth before and after irrigation event was used to estimate net irrigation water, including crop transpiration and soil evaporation. For sprinkling irrigation, the net irrigation water was denoted by the sprinkling discharge, including interception, soil evaporation, and transpiration. The weighted calculated results were representative to show the irrigation water use efficiency at regional scale. Since the measured irrigation data used in this study are not publicly accessible, so sorry for that we are not allowed to share the data.

Tab.C1 Methods for calculating irrigation water use efficiency for irrigation districts

| Direct measurement method | | | | Head-end measurement method |
|---|---|---|---|---|
| Crop type | Dry farming | | $I_{neti} = H \cdot (\theta_{v2} - \theta_{v1})$ | $I_{neti} = \min\left(\dfrac{I_{ini} - D_i}{A_i}, M_i\right)$ |
| | Rice | Submerged irrigation | $I_{neti1} = H_2 - H_1$ | |
| | | Damp irrigation | $I_{neti2} = H \cdot (\theta_{v2} - \theta_{v1})$ | |
| Sprinkling irrigation | | | $I_{neti} = \dfrac{\varepsilon \cdot Q_i}{A_i}$ | |

Net irrigation water of irrigation district: $I_{net} = \dfrac{\sum_i^n I_{neti} \cdot A_i}{\sum A_i}$     Utilization coefficient: $\eta = \dfrac{I_{net}}{I}$

Note: $I_{neti}$ is the net irrigation water amount for each crop estimated from selected typical field patches, mm; $i$ denotes the crop type; $I_{net}$ is the weighted average of net irrigation water for the whole irrigation districts, mm; $I$ is the total irrigation water diverted from water-supplying area for irrigation districts, mm. For dry farming and damp irrigation stage of rice fields, $H$ is the depth of moist soil layer, mm; $\theta_{v1}$ and $\theta_{v2}$ are the soil volumetric water content before and after irrigation events, %. For the submerged irrigation stage of rice fields, $H_1$ and $H_2$ are the depth of water surface before and after irrigation events, mm. In addition, for sprinkling irrigation, $\varepsilon$ is sprinkling efficiency; $Q_i$ is sprinkling discharge, m³; and $A_i$ is the total cover area of crop $i$. When applying the head-end measurement method, the judgement of whether

sufficient irrigation is the premise for accurate estimation of net irrigation water. $I_{ini}$ and $D_i$ are the water amount flowing in and out of fields, mm; $M_i$ is the estimated net irrigation quota for each crop, mm. Notably, rice fields have no drainage during crop growth period.

2. Next, to the data issue I am not sure whether your calculations are correct. According to Eq 4 you calculate the equivalent precipitation as the sum of irrigation (I), precipitation (P) and groundwater evaporation (ETg). But how does this relate to your water balance in Eq 9? Is ETg part of ET? And more importantly: how do you define ET? I think that in Eq 9 ET equals the total actual evaporation (=sum of interception evaporation Ei, transpiration Et and soil evaporation ETg?). However, in Eq 10 it seems that ET is equal to transpiration, as Peffd equals P minus interception. And how is it possible that according to Eq 10 all water entering the unsaturated zone is evaporating? This would mean that no water is percolating to the ground water reservoir? Hence, to summarize, I have some doubts on the water balance closure in relation to how you define evaporation. A schematic conceptual overview might help to clarify this.

**Response:** Thanks for your suggestion. Here, we will clarify the water balance process with the citation of Fig.3 used in our previous study (Chen et al., 2020) (termed as Fig.B2 in Appendix in revised manuscript). For the water balance process in irrigation districts, the water input components include precipitation (P) and irrigation water (I), the output components include evapotranspiration (ET) and runoff (Qout), both of which induce the variation in water storage together ($\triangle$S, including soil water storage and groundwater storage) (Fig.B2 A). Thus, the water balance equation can be expressed as:

$$P + I = ET + Qout + \triangle S \qquad (1)$$

In this study, we followed the perspective to conduct water balance for upper vadose zone only proposed in our previous study (Chen et al., 2020). "While the aquifer was divided into soil layer and groundwater aquifer, the exchange flow between them should be considered as components for water balance analysis". The groundwater recharge (GR) from infiltration of soil layer is supposed to be output component of water balance. "For areas with groundwater depth less than 3 m, the upward movement of groundwater served as input component for water balance together with precipitation and irrigation water (Fig.B2 B). Evapotranspiration and infiltration to groundwater were taken as output components of water balance, together leading to the variation in soil water storage ($\triangle$Si) (Fig.B2 B)." Thus, the water balance equation for upper vadose zone with shallow groundwater can be expressed as:

$$P + I + ETg = ET + GR + Qout + \triangle Si \qquad (2)$$

For the area with groundwater depth larger than 3 m (the phreatic evaporation can be negligible), the water balance equation can be expressed as:

$$P + I = ET + GR + Qout + \triangle Si \qquad (3)$$

While the soil water storage was negligible at annual scale, it's reasonable to regard the sum of precipitation, irrigation water, and groundwater evaporation as water availability for evapotranspiration.

We agree with you the total actual evapotranspiration equals to the sum of interception (Ei), transpiration (T), and soil evaporation (E), i.e., ET=Ei+T+E. For irrigation districts with flood and furrow irrigation, the irrigation water flows along canals and land surface so that there is no interception. For sprinkling irrigation, the interception should be considered. In our study, the total actual evapotranspiration was estimated as the sum of net irrigation water and effective precipitation (equation 10). The net irrigation water was estimated as the product of irrigation water use efficiency and gross irrigation water, both of which were measured and provided by China Irrigation and Drainage Development Centre. In the issued report, the net irrigation water referred to the partition of irrigation water actually used for evapotranspiration. The measurement methods for net irrigation water are shown in Tab.C1 in Appendix. For various crop types (dry farming or rice), the difference of soil water content in moist soil layer or water surface depth before and after irrigation event was used to estimate net irrigation water, including crop transpiration and soil evaporation. For sprinkling irrigation, the net irrigation water was denoted by the sprinkling discharge, including interception, soil evaporation, and transpiration. Sorry for the missing information about sprinkling irrigation, which has been added in Tab.C1 in revised manuscript (Please see Tab.C1 in our response to the comment of Page 2, Number 1 above.).

[Figure]

Fig.B2 Structure of water balance model in areas with deep groundwater (A) and shallow groundwater (B) (Chen et al., 2020)

Besides my major concerns related to the data validity and water balance calculation, the manuscript is well written. It's easy to read in good English, well structured, and the Figures are OK.

Detailed comments:

1. P3 L44: "...used AT global and regional scales...."

**Response:** Thanks for your suggestion, we have changed the preposition to "..used at global and regional scales...." in revised manuscript

2. P4 L70-71: the unit of ET, ET0 and P is mm/y.

**Response:** Thanks for your suggestion, we have added the unit of ET, $ET_0$, and P in revised manuscript.

3. P5 L103-106: are this the only irrigation methods? What about furrow or sprinkling? This would have a large impact on e.g., interception 'losses'

**Response:** Sorry for the missing information about the irrigation methods. We agree with you that the interception is influenced by irrigation methods. The interception induced by irrigation events only occurs for sprinkling irrigation since the irrigation water flows along canals and land surface during flood or furrow irrigation. Actually, the irrigation methods including furrow and sprinkling have been taken into consideration while selecting typical field patches for each crop type in the process of calculating irrigation water use efficiency. For various crop types (dry farming or rice), the difference of soil water content in moist soil layer or water surface depth before and after irrigation event was used to estimate net irrigation water, including crop transpiration and soil evaporation. For sprinkling irrigation, the net irrigation water was denoted by the sprinkling discharge, including interception, soil evaporation, and transpiration. The detailed information about the measurement of net irrigation water with irrigation methods considered is added as follow:

> "Referring to the report released by China Irrigation and Drainage Development Centre, at least three typical field patches for each crop type in every large irrigation district with clear boundary, regular shape and moderate area were selected to calculate net irrigation water, while the crop types, irrigation types, soil texture, irrigation methods, and groundwater depth were taken into account comprehensively."

4. P5 L111: Figure 1A doesn't show meteo data. It shows the aridity index. Similar to my comments on data availability regarding irrigation: is meteo data available?

**Response:** Thanks for your suggestion. In our study, "the meteorological data including precipitation, wind speed, air temperature, and relative humidity from weather stations on or around the selected irrigation districts covering the same period were downloaded from China meteorological data network (http://data.cma.cn/). Penman Monteith equation suggested by FAO56 was used to estimate potential evaporation". In our study, the climate conditions were classified according to the aridity index ($ET_0/P$), and the

analysis results were grouped by climate conditions as well. Therefore, we present the distribution of aridity index in Fig.1A instead of meteorological data to achieve the conciseness of figure structure. The description of Fig.1 was revised as:

"A total of 371 large-sized artesian diversion irrigation districts with designed irrigation area covering from 200 to 10000 km$^2$ across China were selected in this study (Fig.1A). The irrigation districts are grouped by climate conditions (Fig.1A), which are classified as arid, semi-arid, semi-humid and humid areas according to the values of aridity index (Tab.1).

[Figure]

Fig.1 Location of the selected irrigation districts under various aridity conditions (A) and the distribution map of mean values of NDVI (B), proportion of clay (C) and sand (D) in China"

5. Eq2: I highly recommend to use single characters in equation and not to use acronyms like ET. ET can be mathematically confused with E*T. Better use sub- and superscripts.

**Response:** Thanks for your suggestion. In previous studies, ET and $ET_0$ have been widely used to represent evapotranspiration and potential evaporation, respectively. So, we stick to using the symbols of ET and $ET_0$ for readability in our study. To avoid the confusion caused by the use of acronyms, we have introduced the abbreviation for each term at their first appearance in revised manuscript.

6. P6 L120-121: I don't understand this sentence.

**Response:** Sorry for the poorly expressed sentence. In this study, we extracted remote sensing data such as NDVI based on the boundary maps of irrigation districts. For the

irrigation districts without available boundary map, we used the circles of same cover area as irrigated area instead. We have rephrased the sentence as follow:

> "Due to the lack of boundary maps of irrigation districts, the circles of same cover area as irrigation districts were used instead to locate the irrigation districts on map and extract data"

7. Section 2.2: How do you know that the irrigation water is originating from the same catchment. If you have transport external water into your catchment, you are violating the water balance.

8. P8 L149: Why do you add ETg? What does it matter if the plants use water from the unsaturated zone, shallow or deep groundwater?

**Response:** Sorry for the ambiguous descriptions of water balance process and the definition of ETg. We agree with you that "the application of water diversion for irrigation districts has transferred the local natural hydrological processes to a new water balance", so we sticked to using the new perspective to conduct water balance for upper vadose zone only which was proposed in our previous study (Chen et al., 2020). "While the aquifer was divided into soil layer and groundwater aquifer, the exchange flow between them should be considered as components for water balance analysis" (Fig.B2 in Appendix in revised manuscript). The groundwater recharge (GR) from infiltration of soil layer is supposed to be output component of water balance (Fig.B2 A). For the areas with shallow groundwater (depth < 3 m), ETg indicates "the groundwater uptake defined as the upward movement of groundwater induced by active crop growth and strong evaporation", i.e., the upward movement of groundwater, serving as input component for water balance in soil layers (Fig.B2 B). "Evapotranspiration and infiltration to groundwater were taken as output components of water balance, leading to the variation in soil water storage together with other components". The groundwater evaporation was estimated by modified Aver'yanov's phreatic equation. Thus, the crop water consumption from unsaturated zone, shallow or deep groundwater were generalized as components in water balance equation. The application of Budyko hypothesis for irrigation districts was conducted based on the new water balance in which the water availability was estimated as the sum of precipitation, irrigation water, and groundwater evaporation.

[Figure]

Fig.B2 Structure of water balance model in areas with deep groundwater (A) and shallow groundwater (B) (cited from the Fig.3 in the study of Chen et al., 2020)

9. P10 L189-190: How is the net irrigation determined? Is transpiration measured to calculate the Water Use Efficiency? How is transpiration measured?

**Response:** Sorry for the uncompleted information about net irrigation water. In this study, the net irrigation water for irrigation districts was estimated as the product of irrigation water use efficiency and total annual irrigation water, both of which were measured and provided by China Irrigation and Drainage Development Centre (L103-107 in revised manuscript). In the issued report, the net irrigation water referred to the partition of irrigation water actually used for evapotranspiration. To measure the net irrigation water, "at least three typical field patches for each crop type in every large irrigation district with clear boundary, regular shape and moderate area were selected to calculate net irrigation water, while the crop types, irrigation types, soil texture, irrigation methods, and groundwater depth were taken into account comprehensively. The field patches were selected in the control range of delivery canals along upper, middle, and lower reaches, respectively". The detailed measurement methods for net irrigation water are shown in Tab.C1 in Appendix (Please see Tab.C1 in our response to the comment of Page 2, Number 1 above). For various crop types (dry farming or rice), the difference of soil water content in moist soil layer or water surface depth before and after irrigation event was used to estimate net irrigation water, including transpiration and soil evaporation. For sprinkling irrigation, the net irrigation water was denoted by the sprinkling discharge, including interception, soil evaporation, and transpiration. Therefore, transpiration was not measured separately but included by the net irrigation water consumed by evapotranspiration.

10. Eq8: this equation effectively calculates interception. Interception is highly dependent on vegetation type; however, I do not see where vegetation has a role in Eq8.

**Response:** Thanks for your suggestion. According to the Soil Conservation Service of U.S.D.A. (1967), effective rainfall is that which is received during the growing period

of a crop and is available to meet consumptive water requirements. It does not include surface run-off or deep percolation losses (Dastane, 1978). The U.S. Department of Agriculture's Soil Conservation Service has developed a procedure for estimating effective rainfall by processing long term climatic and soil moisture data. Without the detailed site-specified information, however, "the empirical equation proposed by U.S. Department of Agriculture's Soil Conservation Service was suggested to estimate effective precipitation in areas with slope less than 5°, which has been widely used and performed well in numerous studies and crop models including in China (Cao et al., 2014a; Cao et al., 2014b; Qin et al., 2016; Zheng et al., 2020)". We agree with you that the interception is highly dependent on vegetation type. In the future study, we will try to do further research to consider the influence of vegetation type on the estimation of interception with the presence of more detailed data.

11. Eq 9: I think this equation should read: Ei + ET=I+P-D

**Response:** Thanks for your suggestion. We agree with you the evapotranspiration is the sum of interception (Ei), transpiration (T), and soil evaporation (E), i.e., ET=Ei+E+T. For irrigation districts, the actual amount of precipitation falling to the ground is the total precipitation minus interception (P-Ei). The water balance equation is I+P-Ei-D=E+T, which can be transferred into I+P-D=Ei+E+T, i.e., I+P-D=ET.

12. P11 L217: unit RSME is mm/y

**Response:** Thanks for your suggestion. We have revised the unit of RMSE into "mm/yr" in revised manuscript.

13. Fig 2: is the water balanced ET on the y-axis calculated based on Eq 9, and thus includes I? If so, it shows to me the potential errors in the irrigation estimates, as for high MODIS-ET the data points start to deviate from the 1:1. Especially, for arid areas irrigation is important.

**Response:** Thanks for your suggestions. As far as possible, we tend to understand the variation pattern of evapotranspiration (ET) in large irrigation districts and conduct attribution analysis at national scale. Actually, it is extremely difficult to calculate actual evapotranspiration for the national scale. Here we use the water balance method with measured irrigation water to calculate ET, which we think is relatively accurate for the national scale under current conditions. "As shown in Fig.2, the water balance equation performed well in estimating $ET$ under current conditions compared with that of MOD16 product with RMSE of 124.4 mm/yr, MRE of 18.6%, and $R^2$ of 0.6. In terms of national scale, it's reasonable to believe that the estimated results of water balance equation were relatively accurate in the following study". Further research on improving the accuracy of method in estimating ET with the presence of more detailed data will be conducted in future study.

14. Fig 3: why is the x-axis of 'semi-arid' different? And should ETg not be included here? And how is ETg determined?

**Response:** Sorry for the typo in X-axis of semi-arid areas. The water availability was used as Pe, i.e., the sum of irrigation water, precipitation, and groundwater consumption. Only for the areas with groundwater depth more than 3 m, Pe was replaced with (I+P) since the groundwater evaporation was negligible. We have revised $ET_0/(I+P)$ for semi-arid areas into $ET_0/Pe$ in Fig.3 in revised manuscript.

15. Eq 11: 'exp' should not be in italic.

**Response:** Thanks for your suggestion. We have unitalicized 'exp' in equation 11 in revised manuscript.

16. Fig 8b: Are the observed ET values from MODIS? Please note that MODIS is a model.

**Response:** Sorry for the uncompleted description about the titles in Fig.8. The data of Y axis refer to the evapotranspiration calculated by Budyko equation with the use of ω estimated by equation (11). The data of X axis refer to the evapotranspiration estimated by water balance equation shown in equation 10. The comparison tends to validate the accuracy of ω estimated by equation (11). We have rephrased the title of Fig.8 in revised manuscript as follows:

> "Fig.8 A: Comparison between optimal values of ω and estimated ones from multiple linear regression analysis; B: Relationship between water balance ET and Budyko ET with the use of ω calculatd by Equation (11)".

**References:**

Cao, X.C., Wu, P.T., Wang, Y.B. and Zhao, X.N., 2014a. Assessing blue and green water utilisation in wheat production of China from the perspectives of water footprint and total water use. Hydrology and Earth System Sciences, 18(8): 3165-3178.

Cao, X.C., Wu, P.T., Wang, Y.B. and Zhao, X.N., 2014b. Water footprint of grain product in irrigated farmland of China. Water Resources Management, 28(8): 2213-2227.

Chen, H., Huo, Z.L., Zhang, L. and White, I., 2020. New perspective about application of extended Budyko formula in arid irrigation district with shallow groundwater. Journal of Hydrology, 582.

Dastane, N.G., 1978. Effective rainfall in irrigated agriculture. Food and Agriculture Organization of the United Nations, Rome, Italy.

Qin, L., Jin, Y., Duan, P. and He, H., 2016. Field-based experimental water footprint study of sunflower growth in a semi-arid region of China. J Sci Food Agric, 96(9): 3266-73.

Zheng, X.X., Qin, L.J. and He, H.S., 2020. Impacts of Climatic and Agricultural Input Factors on the Water Footprint of Crop Production in Jilin Province, China. Sustainability, 12(17).